# Textile Bandwidth-Enhanced Coupled-Mode Substrate-Integrated Cavity Antenna with Slot

Jie Cui [1,2], Feng-Xue Liu [3,4,5,*], Xiaopeng Shen [6], Lei Zhao [2] and Hongsheng Yin [2,*]

1   School of Transportation Engineering, Jiangsu Vocational Institute of Architectural Technology, Xuzhou 221116, China
2   School of Information and Control Engineering, China University of Mining and Technology, Xuzhou 221116, China
3   School of Physics and Electronic Engineering, Jiangsu Normal University, Xuzhou 221116, China
4   Jiangsu Xiyi Advanced Materials Research Institute of Industrial Technology, Xuzhou 221400, China
5   Jiangsu Normal University Kewen College, Xuzhou 221132, China
6   School of Materials and Physics, China University of Mining and Technology, Xuzhou 221116, China
*   Correspondence: liufengxue@jsnu.edu.cn (F.-X.L.); yhs@cumt.edu.cn (H.Y.)

**Abstract:** A textile bandwidth-enhanced coupled-mode substrate-integrated cavity antenna with a slot is presented. The original coupled-mode substrate-integrated cavity antenna is of two close resonances for the odd and even coupled modes, and a rectangular slot is added on the top layer to introduce a third resonance. Parameters are optimized to merge the bands of the three resonances to realize a widened $-10$ dB impedance band to cover the Medical Body Area Network band, 2.45 GHz Industrial Scientific Medical band and Long-Term Evolution Band7. The proposed antenna can operate in a $-10$ dB impedance band of 2.32–2.69 GHz with a 14.9% fractional bandwidth according to the measurements on a fabricated prototype. Simulation and measurement results illustrate the robustness of the proposed textile antenna in the vicinity of the human body and cylindrical bending conditions. In addition, the simulated specific absorption rate of the antenna radiation in the human body is lower than the IEEE and EN limits.

**Keywords:** wearable antenna; textile antenna; bandwidth enhancement; substrate-integrated cavity antenna; coupled mode





## 1. Introduction

In the last few years, wearable technology has been investigated and widely applied in athlete training [1], medical monitoring [2], military equipment [3], security gear [4] and other areas. It plays an important role in the wireless body area network (WBAN) and is required to have several characteristics for practical applications [5]. First of all, a small profile, a light weight and a high flexibility are required for the comfort of the wearer. Secondly, the antenna should be integratable into clothes, with stable performance on the human body and in bending conditions. Textile antennas are a good candidate to meet such requirements [6–9].

The substrate-integrated waveguide (SIW) antenna [10] has been selected in the literature as a suitable candidate for textile wearable antennas due to its planar configuration with an easy integration into textile materials through embroidery [11,12]. The half-mode substrate-integrated waveguide/cavity (HMSIW/HMSIC) antenna [13,14], developed from the SIW antenna, comprises only half of the 2D footprint of the full SIW antennas, and is more suitable for wearable applications [15]. In certain applications, wearable antennas are required to operate in multiple bands including the Medical Body Area Network (MBAN) band (2.36–2.4 GHz), the 2.45 GHz Industrial Scientific Medical (ISM) band (2.4–2.835 GHz), and the Long-Term Evolution Band7 (LTE-7 band) (2.5–2.57 GHz for upload and 2.62–2.69 GHz for download) [16,17]. However, the basic SIW/HMSIW/HMSIC

antennas are narrow-band due to their highly frequency-selective geometries [18,19]. Therefore, several bandwidth-enhancing strategies for the wearable/flexible SIW antenna or planar inverted-F antenna (PIFA) have been reported. In reference [20], a bandwidth-enhanced HMSIW antenna with a cork substrate was designed for wearable applications. By adding a slot in the top patch of the HMSIW antenna, two hybrid modes combining different proportions of the $TM_{110}$ and $TM_{120}$ modes were excited with close resonance frequencies to obtain an enhanced fractional bandwidth of 23.7% at 5.5 GHz. Reference [17] introduced a wearable SIW antenna with a miniaturized structure composed of two coupled eighth-mode cavities and an enhanced bandwidth of 414 MHz (16.2%) covering the 2.45 GHz ISM band and LTE-7 up- and downlink-bands. Reference [21] presented a wearable dual-band HMSIC antenna with an aperture slot to raise the bandwidth from 4.5% to 6.0% at 2.45 GHz. Reference [22] proposed an all-textile low-profile wearable PIFA with a slot and shorting pins with an enhanced fractional bandwidth of 18% at 5.5 GHz. The added slot and shorting pins, respectively, shifted the resonances of the $TM_{2,1/2}$ and $TM_{0,1/2}$ modes towards each other until two bands were merged into a widened band. Our previous work [23] also introduced two textile HMSIC antennas with added metallic/textile shorting vias to increase the lower resonance frequency towards the higher resonance frequency to realize bandwidth enhancement (14.7) at 5.5 GHz.

The coupled-mode patch antennas (CMPAs) proposed in reference [24–26] also provided inspiration for the bandwidth enhancement of the wearable SIW antenna. These CMPAs were of two resonances, representing the odd and even coupled modes ($TM^{HM}_{0,1,0}$ mode) in their two back-to-back half patches. References [27,28] (our previous works) also introduced two coupled-mode substrate-integrated cavity (CMSIC) antennas with similar back-to-back HMSIC structures and dual-band/mode characteristics. For both the CMPA and CMSIC antenna, the resonance frequencies for the odd and even coupled modes are naturally close to each other because the internal E-fields in each HMSIC follow the same distribution of the $TM^{HM}_{1,1,0}$ mode. It can be assumed that a larger bandwidth can be obtained when two resonance frequencies are close enough.

Based on this assumption, a textile bandwidth-enhanced CMSIC antenna with a slot is designed and presented in this paper. The proposed textile antenna combines the back-to-back HMSIC structure which leads to two resonances for the odd and even coupled modes and the added slot on the top layer which leads to a third resonance, and is of an enhanced −10 dB impedance band obtained by the fusion of three bands to cover the MBAN, 2.45 GHz ISM and LTE-7 bands. Therefore, this work provides a novel bandwidth-enhancing method for the wearable SIW antenna. Parameter analysis of several parameters is carried out through simulations. A prototype is fabricated through computerized embroidery to verify the design. The polarization, gain and efficiency of the proposed antenna across its −10 dB impedance band are studied through simulations and measurements. The performance of the proposed antenna on the human body or in bending conditions is also investigated through simulations and measurements. In addition, the specific absorption rate (SAR) in a human body phantom is simulated to evaluate the influence of the radiation of the proposed antenna on the human body.

## 2. Antenna Topology and Design

### 2.1. Baisc HMSIC and CMSIC Antennas

As shown in Figure 1a, the cavity of the basic HMSIC antenna (Antenna I) is defined by the conductive top layer, ground plane, three vertical sidewalls and a radiation aperture, and filled by the substrate material. A coaxial probe is employed as the feeding to excite electromagnetic field in the cavity. According to the classic cavity model, the top layer, ground plane and sidewalls are considered as perfect electric conductors (PECs), while the radiation aperture acts as a perfect magnetic conductor (PMC). By substituting these

boundary conditions into Maxwell's equations, the fundamental mode that can be excited in the cavity is the $TM^{HM}_{1,1,0}$ mode, and the deduced resonance frequency $f_r$ is [19]:

$$f_r = \frac{c}{2\sqrt{\varepsilon_r}}\sqrt{\frac{1}{4w^2} + \frac{1}{l^2}} \tag{1}$$

where $c$ is the vacuum light speed and $\varepsilon_r$ is the relative dielectric constant of the substrate.

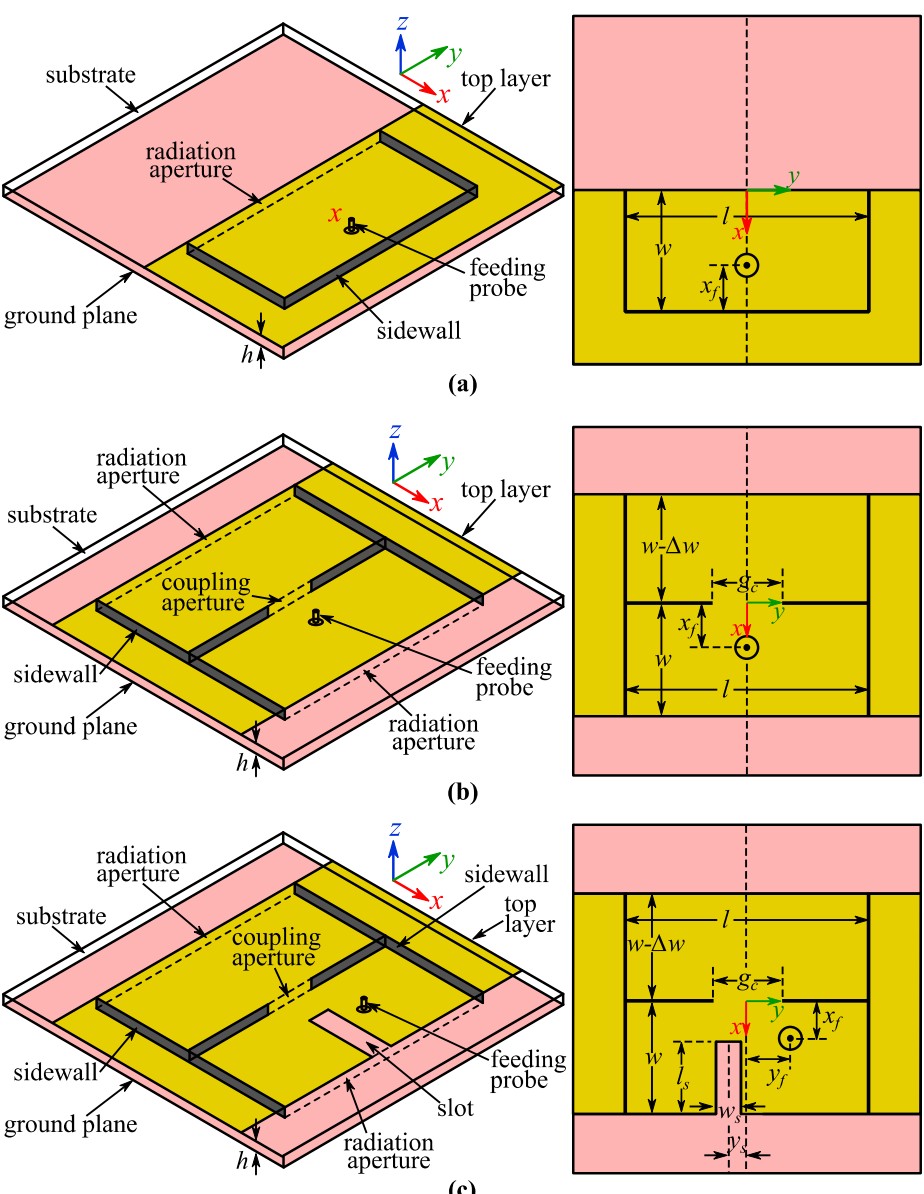

**Figure 1.** Geometries of (**a**) Antenna I, (**b**) Antenna II and (**c**) Antenna III.

As shown in Figure 1b, the basic CMSIC antenna (Antenna II) can be considered as two back-to-back HMSICs with a coupling aperture connecting two cavities. In the lower HMSIC, the fundamental $TM^{HM}_{1,1,0}$ mode can be excited by the feeding probe and coupled into the upper HMSIC through the coupling aperture. The coupling between two HMSICs can be electric or magnetic with different phases between radiation apertures [28].

The employed materials for these two antennas are modelled with the same physical and electrical parameters for comparison. The substrate is selective as the PF-4 foam ($\varepsilon_r = 1.06$, loss tangent $\tan\delta = 0.0001$) with a 3.2 mm thickness ($h$). A 100 mm × 100 mm ground plane is employed to guarantee the isolation between the antenna and the human

body. Both the top layer and ground plane are modelled as planes with a 0.04 Ω/square sheet resistance to imitate the conductive fabric NCS95R-CR. The sidewalls are modelled as vertical planes with a 0.6 Ω/square sheet resistance [28] to imitate the three passes of the linear embroidery of the conductive threads with a 2 mm spacing between the adjacent two stiches. Through simulations, the other dimension parameters for Antenna I and Antenna II are determined as shown in Table 1.

**Table 1.** Dimension parameters of Antenna I, Antenna II Additionally, Antenna III.

|  | $w$ (mm) | $l$ (mm) | $x_f$ (mm) | $y_f$ (mm) | $g_c$ (mm) | $\Delta w$ (mm) | $w_s$ (mm) | $l_s$ (mm) | $y_s$ (mm) |
|---|---|---|---|---|---|---|---|---|---|
| Antenna I | 41.8 | 83.6 | 10.8 | - | - | - | - | - | - |
| Antenna II | 39 | 78 | 11 | - | 24.5 | 2 | - | - | - |
| Antenna III | 42 | 84 | 18 | 13 | 26 | 9 | 7 | 25.5 | 1.5 |

The simulated $|S_{11}|$ curves of Antenna I and Antenna II are presented in Figure 2. For Antenna I, the simulated fundamental mode is obtained at 2.39 GHz with a −10 impedance band of 2.35–2.43 GHz (73.5 MHz bandwidth, 3.1% fractional bandwidth). The simulated band covers the MBAN band, but does not cover other considered bands. For Antenna II, the simulated curve shows two close resonances at 2.39 GHz and 2.49 GHz with a widened –10 dB band of 2.34–2.51 GHz (171.5 MHz bandwidth, 7.1% fractional bandwidth). Compared to Antenna I, the simulated –10 dB impedance band covers the extra frequency band (2.45 GHz ISM band). However, the LTE-7 band is still not covered for Antenna II.

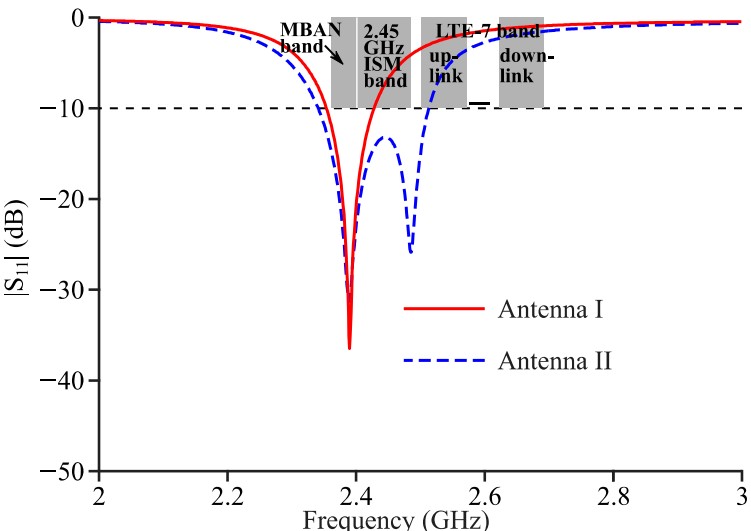

**Figure 2.** Simulated $|S_{11}|$ curves of Antenna I and Antenna II.

### 2.2. CMSIC Antenna with Slot

Based on the geometry of Antenna II, a CMSIC antenna with a slot (Antenna III) is proposed, as shown in Figure 1c. To be specific, a rectangular slot is added on the top layer of the lower HMSIC in Antenna II. The dimension parameters of Antenna III are determined based on simulations as shown in Table 1. The simulated $|S_{11}|$ curve of Antenna III in free space is shown in Figure 3. Three resonances are observed at 2.39, 2.52 and 2.64 GHz. A −10 dB impedance band of 2.33–2.70 GHz with a 374.7 MHz bandwidth (14.9% fractional) is obtained. According to simulation results, all considered frequency bands are well-covered by the enhanced bandwidth of Antenna III.

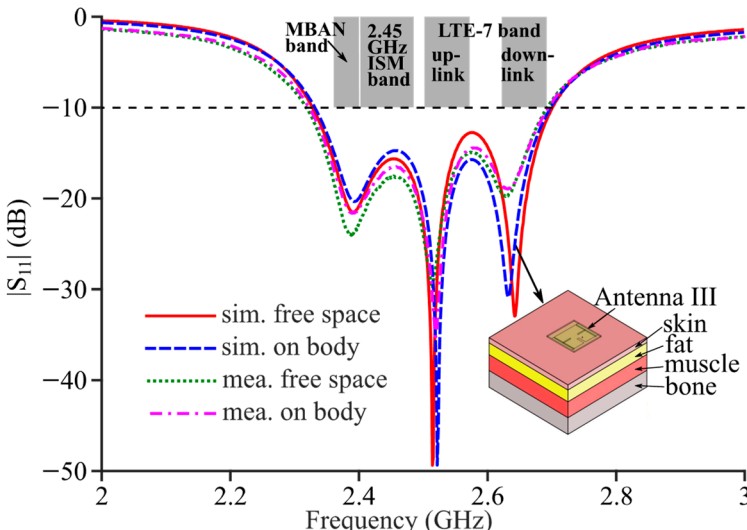

**Figure 3.** Simulated and measured $|S_{11}|$ curves of Antenna III.

The internal vector E-field distributions at the first and second resonance frequencies (respectively denoted as $f_{r1}$ and $f_{r2}$) are simulated and shown in Figure 4 to identify the coupled modes. Both of the simulated E-fields in the lower HMSIC at the first two resonance frequencies follow the distribution for the $TM^{HM}_{1,1,0}$ mode. At $f_{r1}$, the E-fields in two HMSICs are mirror-symmetrical. At $f_{r2}$, the E-fields in two HMSICs are axisymmetric about the *y*-axis. Therefore, it can be recognized that Antenna III is operating with the even and odd coupled modes at $f_{r1}$ and $f_{r2}$, respectively.

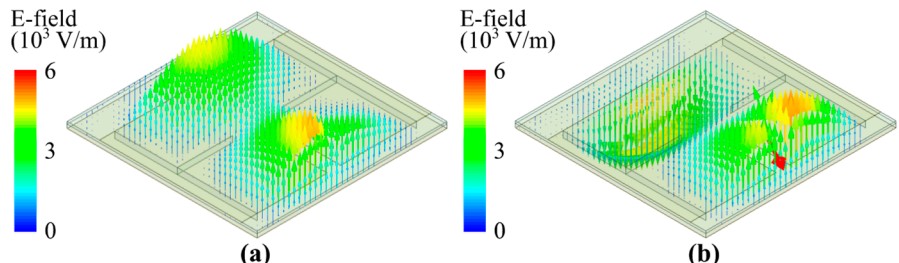

**Figure 4.** Simulated internal vector E-field distributions of Antenna III at (**a**) 2.39 GHz and (**b**) 2.52 GHz.

Figure 5 shows the simulated axial ratio (AR) of Antenna III across 2.33–2.70 GHz, and a 3 dB AR band around 2.60 GHz (2.58–2.62 GHz, 38 MHz bandwidth) is observed. The low AR (6 dB) at the third resonance frequency (denoted as $f_{r3}$) indicates the existence of the circular polarization. Antenna III is of a variable polarization at different frequencies within its −10 dB impedance band. Therefore, its 3 dB AR bandwidth is much smaller than its −10 dB impedance bandwidth.

Figure 6 shows the simulated internal vector E-field distributions of Antenna III at $f_{r3}$. It can be found that the E-field basically only exists in the lower HMSIC. The lower HMSIC can be approximately divided into a main and a parasitic quarter-mode substrate-integrated cavities (QMSICs) because of the added slot, and the $TM^{QM}_{1,1,0}$ mode is excited in the main QMSIC and coupled into the parasitic counterpart through the added slot. Between the E-fields in the main and parasitic QMSICs, there is a phase difference of about 40°. Based on the direction of the fringing E-field, the corresponding equivalent horizontal magnetic current is roughly marked as a red dashed polyline with an arrow for different phases. Both the composite vector **E** (blue solid line with arrow) and **M** (red solid line with arrow) rotate counterclockwise in the *xy* plane when the phase increases from 70° to 290°. This indicates that the circular polarization at $f_{r3}$ is in a right-hand form.

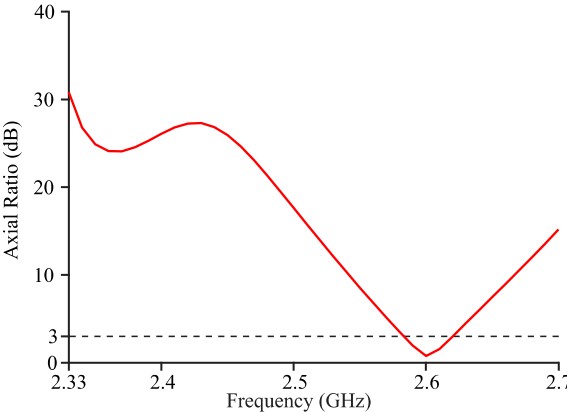

**Figure 5.** Simulated AR of Antenna III across 2.33–2.70 GHz.

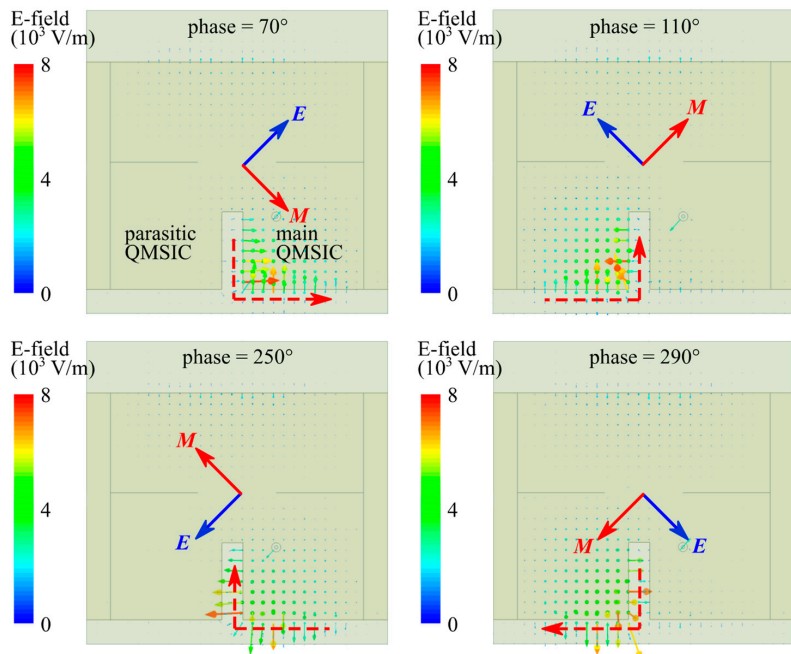

**Figure 6.** Simulated internal vector E-field distributions of Antenna III at 2.64 GHz.

The maximum gain and radiation efficiency of Antenna III in free space across its $-10$ dB band are simulated and presented in Figure 7. In the investigated frequency range, the maximum gain and the radiation efficiency are higher than 5.5 dBi and 86%, respectively. The free-space radiation patterns of Antenna III at its three resonance frequencies are obtained through simulations, as shown in Figure 8. At the first and second investigated frequencies, the coplanar/cross polarized gain is $\text{Gain}_\theta/\text{Gain}_\phi$ in the $xz$ plane and $\text{Gain}_\phi/\text{Gain}_\theta$ in the $yz$ plane. At $f_{r1}$, the maxima of the coplanar and cross polarized gains are 7.6 and $-8.3$ dBi, respectively. At $f_{r2}$, the maxima of the coplanar and cross polarized gains are 8.8 and $-3.8$ dBi, respectively. At the third resonance frequency, however, the far-field radiation can be considered as the sum of the radiations of two parts of the equivalent magnetic current (red dashed polyline with arrow in Figure 6) along the radiation aperture of the lower HMSIC and the added slot. This leads to the increases in $\text{Gain}_\theta/\text{Gain}_\phi$ in the $xz/yz$ planes. Figure 8c shows that the simulated maximum of $\text{Gain}_\theta$ is higher than $\text{Gain}_\phi$ in the $xz$ plane, and the simulated maximum of $\text{Gain}_\phi$ is higher than $\text{Gain}_\theta$ in the $yz$ plane. Therefore, at the third resonance frequency, the coplanar/cross polarized gain is $\text{Gain}_\theta/\text{Gain}_\phi$ in the $xz$ plane and $\text{Gain}_\phi/\text{Gain}_\theta$ in the $yz$ plane, and the maxima of the coplanar and cross polarized gains are 4.7 and 0 dBi, respectively. In addition, the maxima

of the right-hand circular polarized (RHCP) and left-hand circular polarized (LHCP) gains at $f_{r3}$ are 5.3 and $-1.1$ dBi, respectively. This further proves that the circular polarization at 2.64 GHz is in a right-hand form.

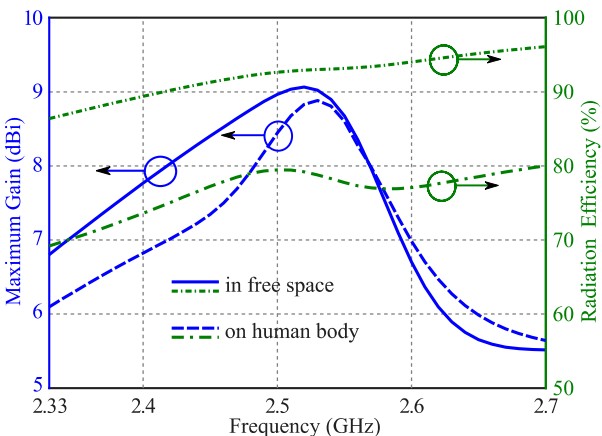

**Figure 7.** Simulated curves of maximum gain and radiation efficiency with respect to the frequency of Antenna III (2.33–2.70 GHz).

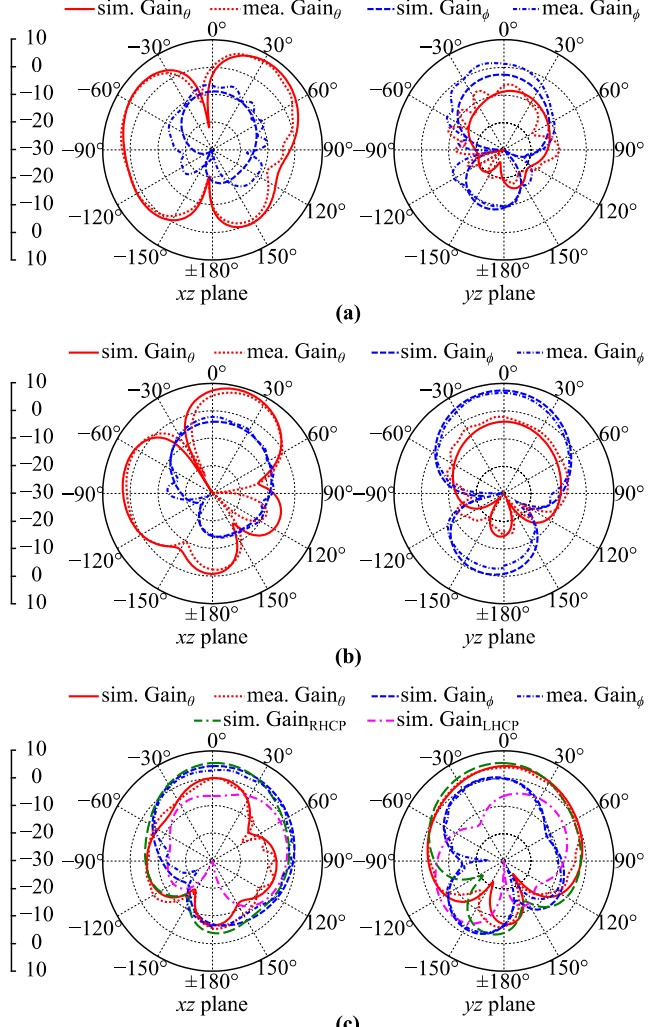

**Figure 8.** Simulated and measured radiation patterns of Antenna III in free space: (**a**) 2.39 GHz; (**b**) 2.52 GHz; (**c**) 2.64 GHz for simulation and 2.63 GHz for measurement.

### 2.3. Parametric Analysis

Parameter $w$ and $l$ are key parameters determining the resonance frequencies of the HMSIC antenna (Antenna I) and the CMSIC antenna (Antenna II), and thus also determine the first two resonance frequencies of Antenna III. Parameter $x_f$ and $y_f$ determine the impedance matching based on the basic knowledge of antenna design. Apart from these parameters, whose influences are well-known, simulations are carried out for Antenna III to study the effects of other parameters, namely $g_c$, $\Delta w$, $l_s$, $w_s$ and $y_s$.

Figure 9a shows the simulated $|S_{11}|$ curves of Antenna III with different lengths of the coupling aperture $g_c$. When $g_c$ increases, the $f_{r2}$ and $f_{r3}$ basically remain unchanged, and the antenna impedance remains well-matched at these two resonances. The first resonance frequency, however, is decreased with an increasing $g_c$, and the −10 dB impedance band is increased.

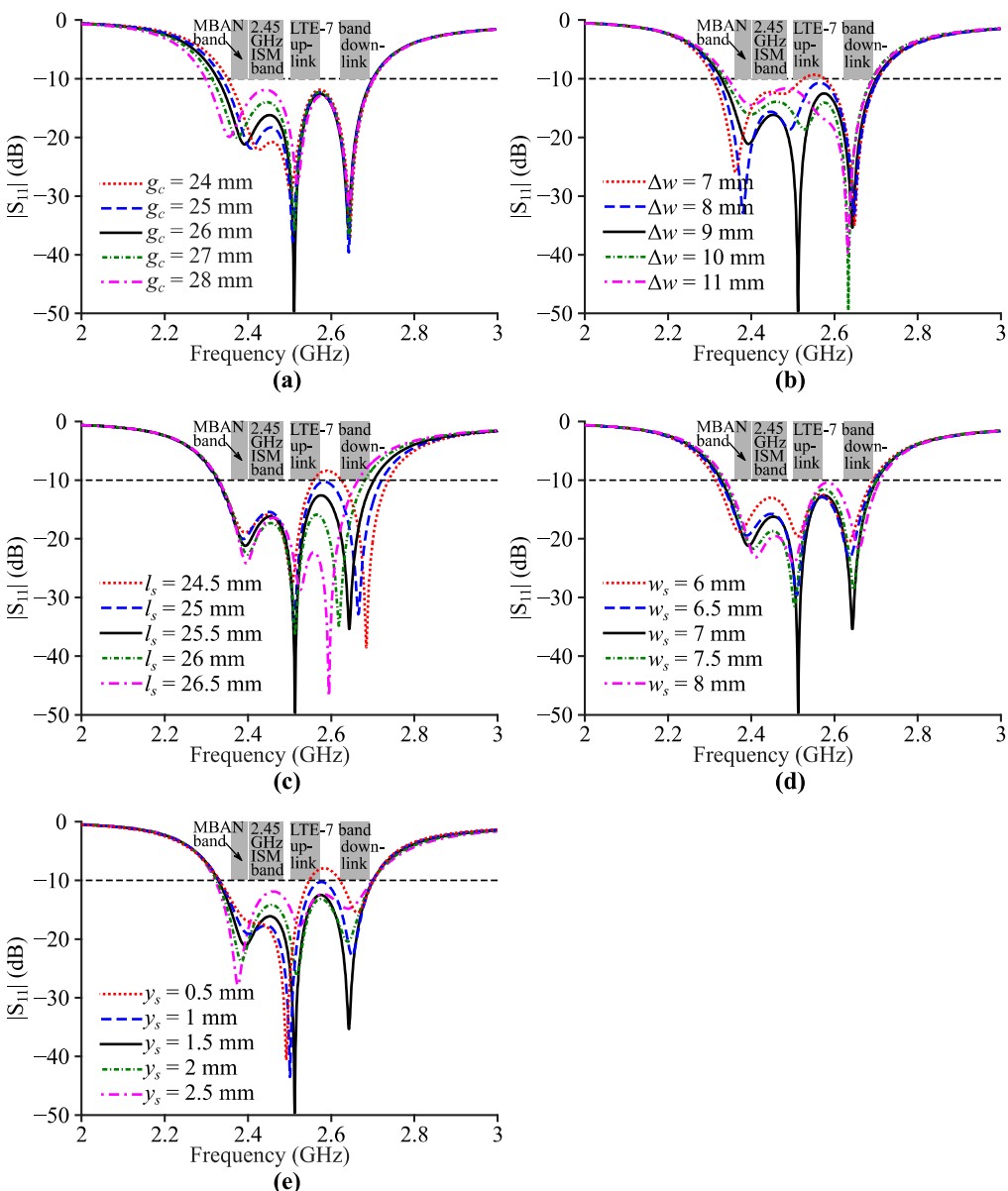

**Figure 9.** Simulated $|S_{11}|$ curves of Antenna III with different values of (**a**) $g_c$, (**b**) $\Delta w$, (**c**) $l_s$, (**d**) $w_s$ and (**e**) $y_s$.

Figure 9b shows the simulated $|S_{11}|$ curves of Antenna III with different values of parameter $\Delta w$. $f_{r3}$ suffers no influence from the change in $\Delta w$ in terms of the frequency and

the impedance matching. With an increasing $\Delta w$, both $f_{r1}$ and $f_{r2}$ increased, and the $-10$ dB impedance bandwidth decreased. This can be explained by the decrease in the cavity size in two HMSICs. In addition, the impedance matching at $f_{r1}$ and $f_{r2}$ is significantly affected by the change in $\Delta w$.

Figure 9c shows the simulated $|S_{11}|$ curves of Antenna III with different lengths of the added slot $l_s$. $f_{r1}$ and $f_{r2}$ are basically not affected by the change in $l_s$ in terms of the frequencies and the impedance matching. When $l_s$ increases, $f_{r3}$ is decreased without deteriorating the impedance matching, and the $-10$ dB impedance band is narrowed. The decrease in $f_{r3}$ can be explained by the increase in the radiating edge for the third resonance due to the increase in $l_s$.

Figure 9d shows the simulated $|S_{11}|$ curves of Antenna III with different widths of the added slot $w_s$. When $w_s$ is increased, $f_{r1}$ and $f_{r3}$ are slightly increased, and the $-10$ dB bandwidth basically stays unchanged. The increases in resonance frequencies can be explained by the shrinking of the effective cavity sizes in the lower HMSIC and the lengths of the radiating edges of the main and parasitic QMSICs due to the increase in $w_s$. $f_{r2}$ basically stays unchanged. In addition, the impedance matching at three resonance frequencies can be affected by the change in $w_s$.

Figure 9e shows the simulated $|S_{11}|$ curves of Antenna III with different values of parameter $y_s$. The change in $y_s$ leads to slight shifts in all three resonance frequencies and noticeable deterioration of the impedance matching at resonance frequencies.

### 2.4. Mutual Effects between Antenna and Human Body

In practical applications, the proposed textile antenna needs to be integrated into clothes and located in the vicinity of the human body. Therefore, it is important to study the mutual influence of the proposed antenna and the human body. A $300 \times 300 \times 40$ mm$^3$ four-layer (skin, fat, muscle and bone) phantom is modelled in simulations to imitate the human body, and the antenna is placed above the top center of the phantom with a 5 mm gap, as shown in the inset of Figure 3. The characteristics and thicknesses for each layer of the phantom at 2.4 GHz are obtained from reference [29], as listed in Table 2.

**Table 2.** Characteristics and thicknesses of each layer of skin, fat, muscle and bone.

| | Relative Permittivity | Conductivity (S/m) | Density (kg/m$^3$) | Thickness (mm) |
|---|---|---|---|---|
| Skin | 38.09 | 1.43 | 1100 | 2 |
| Fat | 5.29 | 0.1 | 910 | 5 |
| Muscle | 52.8 | 1.69 | 1041 | 20 |
| Bone | 11.4 | 0.381 | 1908 | 13 |

The simulated $|S_{11}|$ curve of Antenna III with the human body is shown in Figure 3. Compared to the free-space scenario, $f_{r1}$ and $f_{r2}$ remain unchanged while a slight shift is observed for $f_{r3}$ (2.64 GHz in free space vs. 2.63 GHz on the human body). The $-10$ dB impedance band with the human body is still 2.33–2.70 GHz, and covers all of the considered frequency bands.

The maximum gain and radiation efficiency of Antenna III with the human body across its $-10$ dB band are simulated and presented in Figure 7. In the whole $-10$ dB impedance band, the maximum gain and radiation efficiency are above 5.6 dBi and 69%, respectively. Compared to the scenario without the human body, the radiation efficiency is reduced due to the absorption of radiation by the human body.

The radiation patterns of Antenna III on the human body at its three resonance frequencies are simulated and presented in Figure 10. The definitions of the coplanar and cross-polarized gains at each investigated frequency remain the same as in the free-space scenario. At 2.39 GHz, the maxima of the coplanar and cross polarized gain are 6.7 and $-8.6$ dBi, respectively. At 2.52 GHz, the maxima of the coplanar and cross-polarized gain are 8.6 and $-2.6$ dBi, respectively. At 2.63 GHz, the maxima of the coplanar and cross-

polarized gain are 3.6 and 3.1 dBi, respectively, and the maxima of the RHCP and LHCP gains are 6.3 and −1.2 dBi, respectively. In addition, the backward radiations of all patterns on the human body are dramatically reduced compared to those in free space due to the reflection of the radiation by the human body.

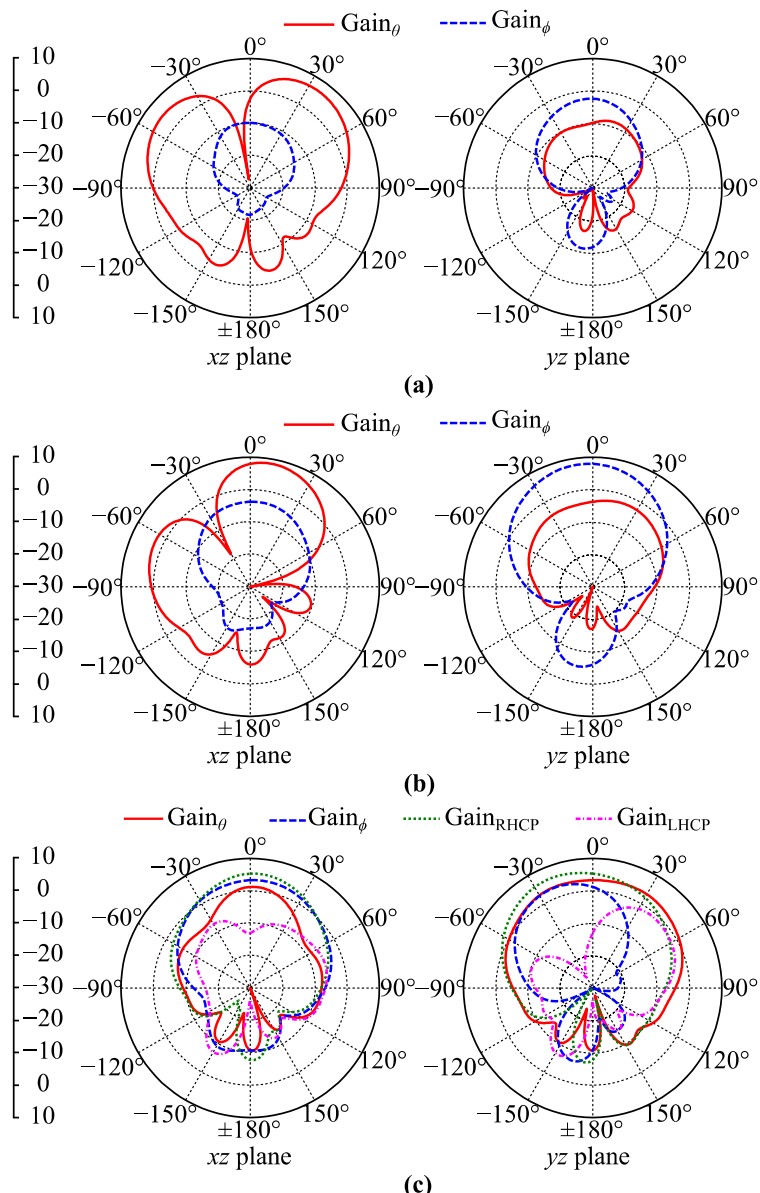

**Figure 10.** Simulated radiation patterns of Antenna III on the human body: (**a**) 2.39 GHz; (**b**) 2.52 GHz; (**c**) 2.63 GHz.

The SAR is simulated and analyzed to investigate the influence of the radiation of Antenna III on the human body. The simulated 1 g average SAR distributions in the four-layer phantom at its three resonance frequencies with a 0.5 W input power are presented in Figure 11. The simulated SARs are below 0.61, 0.55 and 0.86 W/kg at 2.39, 2.52 and 2.63 GHz, respectively, and satisfy the limits of IEEE C95.1-2005 (<1.6 W/kg) and EN 50361-2001 (<2.0 W/kg). When the gap between the human body and the antenna reduces to 1 mm, the corresponding maxima of the SAR increase to 0.79, 0.69 and 1.07 W/kg, which are still well below the mentioned limits. In practical applications, the SAR can be further restricted by employing a larger ground plane, which can be made from conductive fabric

or conductive threads weaved into the garment. Therefore, the safety of Antenna III on the human body is guaranteed.

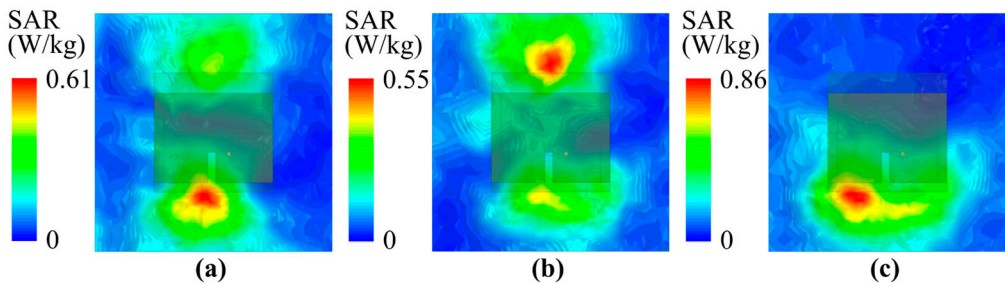

**Figure 11.** Simulated 1 g average SAR of Antenna III in the phantom with a 0.5 W input power: (**a**) 2.39 GHz; (**b**) 2.52 GHz; (**c**) 2.63 GHz.

## 3. Prototype Fabrication and Measured Results

A prototype of Antenna III is fabricated following the procedures listed below. The photos of the fabricated prototype are shown in Figure 12a,b.

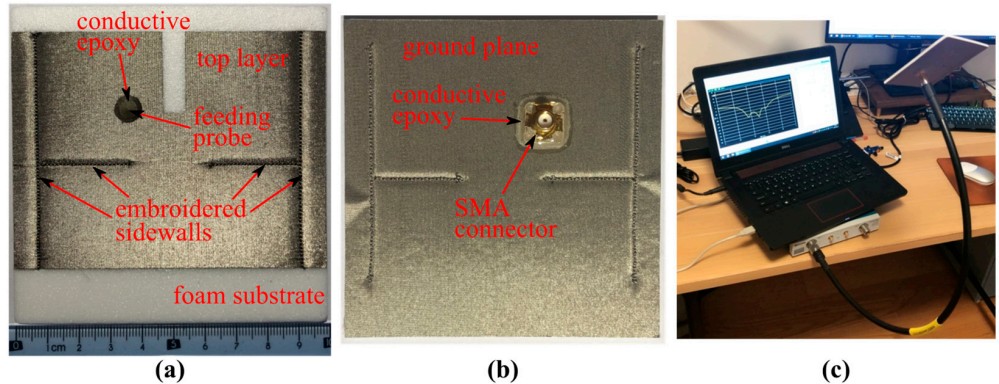

**Figure 12.** Fabricated prototype of Antenna III: (**a**) top view; (**b**) bottom view; (**c**) measurement with a vector network analyzer.

1. Design the pattern for the embroidered sidewalls with the PE-DESIGN software.
2. Cut the PF-4 foam sheet and conductive fabric NCS95R-CR into designed shapes for the substrate, top layer and ground plane, respectively.
3. Glue the top layer and the ground plane onto the top and bottom surface of the substrate.
4. Embroider the sidewalls with the designed pattern using the computerized embroidery machine Brother NV950.
5. Install the SMA connector with conductive epoxy CircuitWorks CW2460.

A vector network analyzer Tektronix TTR506A (as shown in Figure 12c) and an anechoic chamber are used to investigate the $|S_{11}|$ curves and coplanar/cross polarized gain patterns of the prototype. Because of the limited equipment, the radiation efficiency, AR curves, AR bandwidth, RHCP/LHCP gain patterns and SAR on the human body are not measured and presented in this paper.

The measured $|S_{11}|$ curves of Antenna III with/without the human body are shown in Figure 3. The measured $|S_{11}|$ curves show a high agreement with simulated curves in terms of the resonance frequencies and the $-10$ dB impedance bands. In free space, the measured resonances are obtained at 2.39, 2.51 and 2.63 GHz. On the human body, the measured resonance frequencies are 2.39, 2.52 and 2.63 GHz. The measure $-10$ dB impedance band is 2.32–2.69 GHz (14.9% fractional bandwidth) for both free-space and on-body scenarios, and all of the considered frequency bands can be covered.

The $|S_{11}|$ curves of Antenna III with cylindrical bending conditions are also obtained through measurements and are shown in Figure 13. A polyvinyl chloride (PVC) frame

is fabricated to provide a cylindrical surface with a 5.5 cm radius to imitate the situation in which the antenna is fixed on the clothes and bent around the limbs or the shoulder. During measurements, the prototype is fixed onto the PVC frame using paper tapes as shown in the insets of Figure 13. With a bending condition around the *x*-axis, the measured resonance frequencies are 2.39, 2.49 and 2.60 GHz, and the measured −10 dB impedance band is 2.27–2.69 GHz, which still covers the MBAN, 2.45 GHz ISM and LTE-7 bands. With a bending condition around the *y*-axis, however, the impedance matching become worse at the $f_{r2}$ and $f_{r3}$, and the measured −10 dB band shrinks to 2.30–2.66 GHz which does not fully cover the LTE-7 band. This measured result contributes to the strategy of location and direction of the proposed textile antenna on the human body.

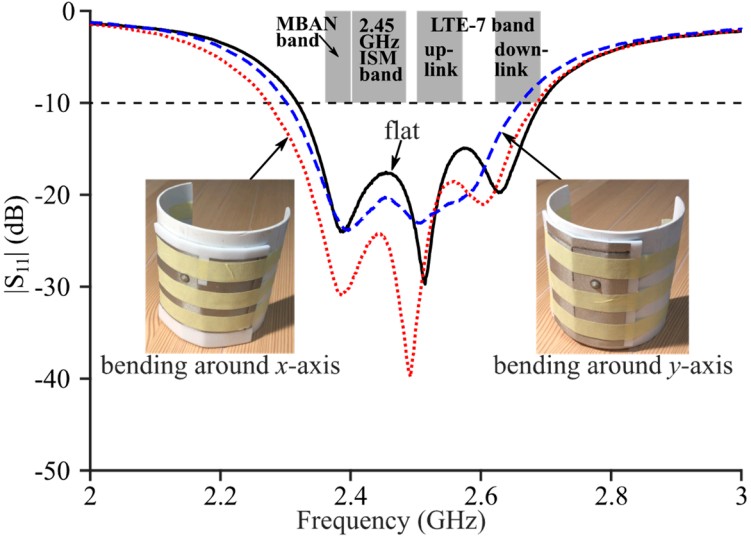

**Figure 13.** Measured |$S_{11}$| curves of Antenna III with/without cylindrical bending conditions.

The measured patterns of the coplanar and cross polarized gain of Antenna III in free space at its resonance frequencies are shown in Figure 8. The measured coplanar polarized gain patterns are in good agreement with the simulated patterns. The maxima of the measured coplanar polarized gain at 2.39, 2.51 and 2.63 GHz are 7.0, 7.6 and 4.0 dBi, respectively. The minor differences between the simulated and measured |$S_{11}$| curves and maxima of the coplanar polarized gain can be explained by slight fabrication/measurement inaccuracies. Unfortunately, the radiation efficiency and the patterns of the LHCP/RHCP gains are not measured in this work due to the limited equipment.

### 4. Conclusions

A textile bandwidth-enhanced CMSIC antenna with a slot has been designed and fabricated to cover the MBAN, 2.45 GHz ISM and LTE-7 bands in wearable applications. Based on the basic CMSIC antenna with two close resonances, a third resonance is introduced by the added slot on the top layer, and parameters are optimized to merge three resonances into a wide −10 dB impedance band. The simulated −10 dB impedance band of the proposed antenna is 2.33–2.70 GHz, satisfying the design requirements. Simulations of the internal vector E-field, AR and radiation patterns indicate that the proposed antenna is linearly polarized at the first and second resonance frequencies and right-hand circularly polarized at 2.64 GHz. Additionally, the maximum gain and radiation efficiency of the proposed antenna in free space are above 5.5 dBi and 86%, respectively, across the −10 dB impedance band based on simulation results. Simulations of the proposed antenna on a four-layer human body phantom prove its stable coverage over the considered frequency bands and safety on the human body. A highly flexible textile prototype of the proposed antenna is fabricated using computerized embroidery. The measured results of the prototype show a satisfactory agreement with simulation results and verify the design. In

addition, the cylindrical bending conditions in two different directions are investigated for the proposed antenna through practical measurements.

The comparison between the proposed antenna and other flexible bandwidth-enhanced SIW antennas and PIFAs is shown in Table 3. The bandwidth of the proposed antenna is lower than [17,20,22] because of its low-loss PF-4 foam substrate [30]. However, the lower overall loss of the proposed antenna also leads to the higher radiation efficiency and maximum gain. The back-to-back HMSIC structure of the proposed antenna leads to a mediocre cavity size among all candidates. However, this work provides a new solution for the bandwidth enhancement on the wearable SIW antennas by employing the CMSIC structure. In addition, the proposed antenna shows variable radiation patterns and polarizations at different frequencies in the $-10$ dB band, and thus can be potentially reused for multiple applications in on-body/off-body WBAN communications.

**Table 3.** Comparison with other wearable/flexible bandwidth-enhanced SIW antennas and PIFAs.

| | Freq. (GHz) | Bandwidth (%) | $Gain_{max}$ (dBi) | Efficiency (%) | Cavity Size ($\lambda_\varepsilon^2$) |
|---|---|---|---|---|---|
| [20] | 5.5 | 23.7 | 4.3 | 85 | 0.364 |
| [17] | 2.45 | 16.2 | 4.7 | 67~82 | 0.096 |
| [21] | 2.45 | 6 | not given | not given | 0.220 |
| [22] | 5.5 | 18 | 5.9 | 74.1 | 0.662 |
| [23] | 5.5 | 14.7 | 8.8 | $\geq$95 | 1.073 |
| This work | 2.5 | 14.9 | 5.5~9.1 | 86~96 | 0.466 |

**Author Contributions:** Conceptualization, J.C.; methodology, H.Y.; software, J.C.; validation, F.-X.L.; formal analysis, J.C.; investigation, J.C.; resources, L.Z. and X.S.; data curation, J.C.; writing—original draft preparation, J.C.; writing—review and editing, F.-X.L.; visualization, J.C.; supervision, H.Y.; project administration, H.Y.; funding acquisition, H.Y. and F.-X.L. All authors have read and agreed to the published version of the manuscript.

**Funding:** This research is funded by Natural Science Foundation of Jiangsu Province, grant number BK20221226, Jiangsu College and University Natural Science Research Project, grant number 18KJB510014, 21KJB510030 and Xuzhou Science and Technology Project, grant number KC19002.

**Data Availability Statement:** All data are included within the manuscript.

**Conflicts of Interest:** The authors declare no conflict of interest.

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
