# Peer review of "Textile Bandwidth-Enhanced Coupled-Mode Substrate-Integrated Cavity Antenna with Slot"

_electronics, doi:10.3390/electronics11152454_

Round 1

Reviewer 1 Report

This manuscript presents a textile bandwidth-enhanced coupled-mode substrate-integrated cavity antenna with slot. The work is interesting and is well presented. Following comments would be helpful to further improve the manuscript.

The abbreviations should be defined at their first occurrence. Most of the abbreviations are not defined at their first occurrence and are defined later in text.

The distance between the human body and the antenna is chosen as 5mm, how this distance has been chosen? Is there any impact on performance with varying distance between the antenna and the human body?

The -10dB impedance bandwidth is mentioned in lines 145-146 for antenna III. Mention the 3dB AR bandwidth in text for antenna III as shown in results (Figure 6).

The -10dB impedance bandwidth shown in Figure 3 and the 3dB AR bandwidth shown is Figure 6 are different. Authors should add explanation to address the variations.

Some of the units are not properly generated in pdf, could be due to compilation error. For example, see first paragraph of page 4. Ohm/??. Please carefully check the manuscript.

Reviewer 2 Report

Hello,

The work presents a textile antenna with slot technique for 2.4 GHz wireless application. The paper has been organized well and depicted good results in both simulation and measurement. In my opinion, it will be considered for publication in the ELECTRONICS-MDPI after a revision.

1.      Why have you plotted simulated and measured co-polarized and cross-polarized gain patterns for antenna-III in two figures (Fig. 8 and Fig. 14)? Please plot both in one figure for better comparison between sim. and meas. results.

2.      For fig10, can you include a field measurement on the body to illustrate that the simulated sample on the body matches the reality?

3.      And also for thermal analysis, can you put it on the body, for example, and test the amount of temperature increase on the body while working and sending information from the antenna?

(For systems that are installed on the body, this thermal test depicts something interesting and important in my opinion that the temperature does not rise during operation to cause burns)

4.      Please add a table for comparison the proposed work with other related papers and please discuss about positive and negative points of the presented work.

5.      Fig.6 shows a CP for the proposed antenna at 2.6 GHz, please describe more in details in the text, it is very favourable.

6.      Please describe about contribution/novelty of the proposed paper in the text.

Taken as a whole, the paper is written in a good style. After a revision it can be considered for publication./Thanks.

Best Wishes.

Reviewer 3 Report

A textile bandwidth-enhanced coupled-mode substrate-integrated cavity antenna with slot is presented in this paper . Here are my comments

1-      The novelty of the work should be added in the introduction section.

2-      The introduction section should be improved and re-written. Many references should be added in introduction section to show the previous state of the arts.

3-      How the CP is generated in your design also how could you enlarge the AR bandwidth.

4-      Why the co and cross polarization levels at y-z plane have the same trend please add discussion.   

5-      Antenna design procedures should be added.

6-      SAR results on human phantom should be added

7-      please add the measured results of the gain and efficiency

8-      The VNA screen should be added with the results

9-      Figures should be discussed in more detail

10-   Table of comparison should be added

11-   New references from 2021-2022 should be added and compared with the author's design.

Round 2

Reviewer 1 Report

Authors have addressed the comments and suggestions. The paper has been improved.

Reviewer 3 Report

All of my concerns are properly addressed.